# Immunotherapy for Colorectal Cancer: Mechanisms and Predictive Biomarkers

**DOI:** 10.3390/cancers14041028

**Published:** 2022-02-17

**Authors:** Lindsey Carlsen, Kelsey E. Huntington, Wafik S. El-Deiry

**Affiliations:** 1Laboratory of Translational Oncology and Experimental Cancer Therapeutics, The Warren Alpert Medical School, Brown University, Providence, RI 02903, USA; lindsey_carlsen@brown.edu (L.C.); kelsey_huntington@brown.edu (K.E.H.); 2The Joint Program in Cancer Biology, Brown University and the Lifespan Health System, Providence, RI 02903, USA; 3Department of Pathology and Laboratory Medicine, The Warren Alpert Medical School, Brown University, Providence, RI 02903, USA; 4Pathobiology Graduate Program, The Warren Alpert Medical School, Brown University, Providence, RI 02903, USA; 5Legorreta Cancer Center, The Warren Alpert Medical School, Brown University, Providence, RI 02903, USA; 6Hematology-Oncology Division, Department of Medicine, Rhode Island Hospital and Brown University, Providence, RI 02903, USA

**Keywords:** colorectal cancer, immunotherapy, checkpoint blockade, adoptive cell therapy, monoclonal antibodies, oncolytic viruses, anti-cancer vaccines, cytokine, T cell, NK cell

## Abstract

**Simple Summary:**

Late-stage colorectal cancer treatment often involves chemotherapy and radiation that can cause dose-limiting toxicity, and therefore there is great interest in developing targeted therapies for this disease. Immunotherapy is a targeted therapy that uses peptides, cells, antibodies, viruses, or small molecules to engage or train the immune system to kill cancer. Here, we discuss the preclinical and clinical development of immunotherapy for treatment of colorectal cancer and provide an overview of predictive biomarkers for such treatments. We also consider open questions including optimal combination treatments and sensitization of colorectal cancer patients with proficient mismatch repair enzymes.

**Abstract:**

Though early-stage colorectal cancer has a high 5 year survival rate of 65–92% depending on the specific stage, this probability drops to 13% after the cancer metastasizes. Frontline treatments for colorectal cancer such as chemotherapy and radiation often produce dose-limiting toxicities in patients and acquired resistance in cancer cells. Additional targeted treatments are needed to improve patient outcomes and quality of life. Immunotherapy involves treatment with peptides, cells, antibodies, viruses, or small molecules to engage or train the immune system to kill cancer cells. Preclinical and clinical investigations of immunotherapy for treatment of colorectal cancer including immune checkpoint blockade, adoptive cell therapy, monoclonal antibodies, oncolytic viruses, anti-cancer vaccines, and immune system modulators have been promising, but demonstrate limitations for patients with proficient mismatch repair enzymes. In this review, we discuss preclinical and clinical studies investigating immunotherapy for treatment of colorectal cancer and predictive biomarkers for response to these treatments. We also consider open questions including optimal combination treatments to maximize efficacy, minimize toxicity, and prevent acquired resistance and approaches to sensitize mismatch repair-proficient patients to immunotherapy.

## 1. Introduction

Colorectal cancer (CRC) is a deadly disease with a 5 year survival rate of 13% when metastatic. Among all types of cancer, CRC is the third deadliest and the incidence of this disease among young people is rising in developing nations [1]. Treatment options for CRC include surgery, chemotherapy, radiation, targeted therapy, and immunotherapy (IT). Chemotherapy regimens generally include 5-fluorouracil (5-FU) as a single agent or in combination with either oxaliplatin, irinotecan, or both. 5-FU or capecitabine (prodrug of 5-FU) combined with oxaliplatin is commonly given to stage II and III CRC patients, while 5-FU, oxaliplatin, and irinotecan are generally administered to stage IV patients. Radiation is most commonly given to stage II–IV rectal cancer patients. Chemotherapy and radiation can elicit severe toxicities in patients, and IT is a promising strategy that aims to kill cancer cells more selectively compared to these traditional treatment options [2].

CRC is classified based on genetic alterations present in the tumor such as mutations in APC, KRAS, BRAF, p53, and mismatch repair (MMR) enzymes [3]. Likely the most relevant genetic alterations in terms of response to IT are those in MMR enzymes. MMR enzymes repair DNA defects, and mutations that lead to MMR deficiency (dMMR) cause an accumulation of mutations at microsatellite regions (also known as short tandem repeats) [4]. Patients with dMMR tumors are classified as microsatellite instable (MSI). MSI tumors can also be referred to as dMMR, MSI+, or MSI-high (MSI-H), whereas patients with proficient MMR enzymes (pMMR) are referred to as microsatellite stable (MSS). MSI status is generally independent of pathological stage. The accumulation of mutations in MSI-H tumors results in the presentation of a large repertoire of neoantigens, commonly referred to as high neoantigen burden [5]. Often, high neoantigen burden improves response to IT in patients with MSI-H tumors as compared to patients with MSS tumors. Some clinical trials have reported benefit of IT exclusively in patients with MSI-H tumors [2]. MSI-H is detected in only approximately 15% of CRC tumors [6], and thus IT is generally ineffective for the majority of CRC patients. Uncovering ways to sensitize MSS CRC to IT is a major area of interest.

IT includes a wide variety of different therapies that engage or train the immune system to kill cancer cells. Different types of IT include immune checkpoint blockade (ICB), adoptive cell therapies (ACT), monoclonal antibodies (mAbs), oncolytic viruses, anti-cancer vaccines, and immune system modulators. Numerous preclinical studies have demonstrated promising results and have been translated into clinical trials. While many of these trials are ongoing, some ITs are currently approved for CRC including mAbs that target cancer-associated antigens, namely panitumumab (anti-EGFR), cetuximab (anti-EGFR), and bevacizumab (anti-VEGF) [7]. Several ICB therapies are also FDA-approved. The mAb pembrolizumab (anti-PD-1) was approved in May 2017 based on results from the KEYNOTE-028 trial, which resulted in a 40% objective response rate in patients with MSI-H tumors [8]. The mAbs nivolumab (anti-PD-1) and ipilimumab (anti-CTLA-4) are also FDA-approved for patients with MSI-H metastatic CRC (mCRC) [9,10]. The clinical success of ICB therapy in specific CRC patient populations has made it highly relevant in the field of cancer IT. It is of great importance to consider combinations of ICB with other types of IT to improve responses.

Additional work has been performed as far as biomarker development to identify groups of CRC patients who are likely to benefit from IT. Promising biomarkers have been identified and are based on specific genetic alterations present in CRC tumors, the CRC tumor microenvironment (TME), and analytes that are detectable within patient serum (i.e., liquid biomarkers). This review will discuss the molecular bases of ITs that have been investigated preclinically and clinically for treatment of CRC, provide an overview of current predictive biomarkers for IT in CRC, will consider future requisite topics of investigation including identification of optimal combination treatments and investigation of methods to sensitize MSS CRC to IT.

## 2. Colorectal Cancer Response to Immunotherapy

The low 5 year survival rate (13% for mCRC) [1], development of resistance to standard therapies, and dose-limiting side effects of cytotoxic drug treatments have inspired the investigation and development of more targeted therapies for treatment of CRC. IT is a promising candidate for treatment of mCRC, especially for patients with MSI-H tumors [2]. In this section, we detail the molecular mechanisms of ITs that have been investigated clinically or preclinically for treatment of CRC. The molecular action and cellular expression of relevant targets are listed in Table 1.

### 2.1. Immune Checkpoint Blockade

Under normal conditions, the body relies on immunosuppressive mechanisms to maintain immune homeostasis and self-tolerance [11,12]. These immunosuppressive mechanisms involve expression of checkpoint proteins, activation of cell death programs, and accumulation of various immunosuppressive cells. While keeping harmful immune reactions in check, these mechanisms can also be exploited by cancer cells to escape immune attack [53]. For example, immunosuppressive checkpoint proteins such as PD-1, CTLA-4, LAG-3, TIM-3, and NKG2A/B and are often upregulated in cancer [54]. These molecules are expressed on the surface of immune cells and, when bound by their cognate ligand, activate immunosuppressive pathways that initiate apoptosis, reduce proliferation, and/or inhibit cytokine secretion in immune cells (Figure 1). In general, ICB therapies block this inhibitory receptor binding and activate anti-tumor immune cells to cause tumor regression. The most thoroughly studied ICB therapies are anti-PD-1/PD-L1 and anti-CTLA-4. These ICB therapies have demonstrated efficacy in the clinic for CRC patients with MSI-H tumors. The clinical success of ICB has given it great relevance in the field of cancer IT and has led to the development of several new directions, including the investigation of other ICBs such as LAG-3, TIM-3, and NKG2. As with all types of IT, ICB has certain advantages and limitations (Table 2). One limitation is systemic toxicity in patients, which most commonly affects organ systems including the skin (30–50%), the gastrointestinal system (20–40%), and the endocrine system (10–50%). These side effects generally include low-grade rashes, diarrhea, and endocrine changes that can be permanent [55]. In addition to these side effects, one main disadvantage of ICB is that most are formulated as mAbs, which sometimes have poor pharmacokinetic properties and tissue accessibility. Formulation of small molecules or modifying mAbs to be smaller could solve these issues. The last main limitation of ICB is that overall response rates in CRC have been low, particularly in the MSS patient population. Despite these limitations, ICB therapy is sensitive, specific, and the wide variety of ICBs available allow combination treatments to target multiple mechanisms, which has the potential to improve efficacy [56]. In the following subsections, we detail the fundamental mechanisms driving the anti-cancer effects of clinically relevant ICBs. We also highlight areas of requisite future investigation to improve response to ICB in CRC.

#### 2.1.1. Anti-PD-1 and Anti-PD-L1

Programmed cell death protein 1 (PD-1) is an inhibitory receptor expressed on CD8+ and CD4+ T cells. When bound by ligand, PD-1 modulates the immune response and promotes self-tolerance by activating apoptosis in antigen-specific T cells and inhibiting apoptosis of regulatory T cells (T regs) [11,12]. PD-1 is bound by programmed cell death ligand 1 (PD-L1), a transmembrane protein that is often overexpressed on tumor cells [12]. Promising in vitro and in vivo preclinical studies have directed clinical trials involving PD-1/PD-L1 inhibition, ultimately resulting in the approval of several anti-PD-1 mAbs.

Clinical use of ICB therapies currently involves the FDA-approved compounds pembrolizumab, nivolumab, and ipilimumab, which can be administered to patients with MSI-H mCRC tumors [57,58]. Several months after the approval of pembrolizumab in May 2017, nivolumab was approved based on results from the CheckMate 142 trial, which demonstrated a 31.1% objective response rate in patients with MSI-H tumors. Nivolumab combined with the anti-CTLA inhibitor ipilimumab was approved in July 2018 after further analysis of the CheckMate 142 trial revealed that patients in this combination treatment group had a 55% objective response rate [8]. Nivolumab can be given alone or in combination with ipilimumab to patients who have progressed after chemotherapy. In 2020, pembrolizumab was approved as a first-line treatment for patients with MSI-H CRC tumors based on the results from the KEYNOTE-177 clinical trial, which reported a median progression-free survival (PFS) of 16.5 months for MSI-H CRC patients treated with first-line pembrolizumab and 8.2 months for patients treated with chemotherapy (HR 0.60; 95% CI, 0.45–0.80; *p* = 0.0002) [59,60]. Recent evidence supports the use of ICB as a frontline treatment. In addition to positive results from the KEYNOTE-177 trial, a phase II trial involving neoadjuvant treatment of MSI-H CRC patients with the anti-PD-1 mAb toripalimab caused a high pathological complete response rate and an acceptable safety profile [61]. In general, anti-PD-1 therapy causes a slightly increased rate of side effects compared to anti-PD-L1 therapy [55].

There are several ongoing clinical trials evaluating the use of ICB in the adjuvant setting. These include evaluation of the PD-L1 inhibitor atezolizumab plus FOLFOX compared to FOLFOX alone in MSI-H CRC patients [62], and evaluation of the anti-PD-L1 compound avelumab following 5-FU-based treatment compared to 5-FU-based treatment alone in patients with stage III MSI-H or *POLE*-mutant CRC [63]. Other clinical trials have tested combination therapies that might improve ICB efficacy in MSS CRC patients with varying results. Combinations of regorafenib (tyrosine kinase inhibitor) and nivolumab initially revealed promising results [64], but this treatment resulted in only modest clinical activity in patients with MSS CRC [65]. A phase II study of pembrolizumab plus capecitabine and bevacizumab revealed an acceptable safety profile and some level of efficacy in MSS CRC patients who have received prior fluoropyrimidine-based therapy. [66].

Recent in vivo investigation points to low levels of tumor-infiltrating dendritic cells (DCs) as a reason for limited response to anti-PD-1/PD-L1 mAbs in MSS tumors. In this study, subcutaneous injection of MSS CRC cells in mice led to the development of tumors that were responsive to ICB whereas orthotopic injection of CRC cells into the liver resulted in tumors that were resistant to ICB. Comparing these tumors revealed that the resistant liver tumors had a significantly lower number of DCs compared to the sensitive subcutaneous tumors. A similar observation was made in liver metastases of CRC patients. As DCs are required for antigen presentation and activation of CD8+ T cells, their low numbers likely contribute to the resistance of liver mCRC cell tumors. This points to combination treatments with ICB and Flt-3 Ligand (FLT3L), a cytokine that promotes the differentiation and/or expansion of immature DCs [67,68]. Further investigation of this nature is needed to determine optimal drug combinations to enhance efficacy of ICB.

#### 2.1.2. Anti-CTLA-4

Cytotoxic T-lymphocyte-associated protein 4 (CTLA-4) is an inhibitory receptor expressed primarily by CD4+ and CD8+ T cells. Antigen-presenting cells (APCs) and cancer cells express the CTLA-4 ligands CD80 and CD86. These ligands can bind to either CTLA-4 or CD28, resulting in a co-inhibitory or costimulatory response, respectively. Binding of CTLA-4 to CD80 or CD86 results in either cell-intrinsic (affecting the T cell) or cell-extrinsic (affecting the secondary cell) immunosuppressive signaling, though most evidence suggests that the CTLA-4 pathway functions primarily through cell-extrinsic mechanisms [13]. CLTA-4 is also expressed on regulatory T cells (T regs, FOXP3+). When CTLA-4 on T regs binds to CD80/86 on APCs, priming and activation of naïve CD4+ T cells by APCs is blocked [69]. In CRC, the anti-CTLA-4 mAb ipilimumab combined with nivolumab is approved for patients with MSI-H mCRC [57]. Strikingly, this combination in the neoadjuvant setting for treatment of early-stage colon cancer resulted in a 100% pathological response in patients with MSI-H tumors and a 27% response rate in MSS tumors after only ~4 weeks of treatment [70]. ICB-related toxicity occurs more frequently in patients receiving anti-CTLA-4 therapy compared to patients receiving anti-PD-1/PD-L1 therapies [55]. Similar to anti-PD-1/PD-L1 therapies, optimal treatment timing and therapy combinations need to be optimized to improve response rates to anti-CTLA-4 ICB therapy, especially in patients with MSS tumors.

#### 2.1.3. Anti-LAG-3

Lymphocyte activation gene-3 (LAG-3) is an inhibitory receptor that is expressed by CD4+ and CD8+ T cells, T regs, and NK cells. LAG-3 on CD4+ cells likely interacts with MHC class II on antigen-presenting cells, while LAG-3 on CD8+ T cells and NK cells likely interacts with L-SECtin on cancer cells. These interactions inhibit immune activity by suppressing T cell proliferation, cytokine production, and Ca2+ flux. LAG-3 on T regs stimulates T reg-mediated immune suppression. The precise mechanisms by which LAG-3 mediates these effects remain to be clarified [14], an important topic of future investigation if optimal treatment combinations are to be established. Several phase II studies are underway to investigate the anti-PD-1 mAbs nivolumab or dostarlimab in combination with other therapies including an anti-LAG-3 antibody [71,72,73]. One trial will specifically investigate responses in patients with MSS tumors [72], the results of which are highly anticipated due to the inefficacy of anti-PD-1/PD-L1 and anti-CTLA-4 ICBs for this patient population.

#### 2.1.4. Anti-TIM-3

T cell immunoglobulin- and mucin-domain-containing-3 (TIM-3), also known as hepatitis A virus cellular receptor 2 (HAVCR2), is expressed on IFN-γ-producing T cells, T reg cells, NK cells, and APCs such as macrophages and DCs. In CD8+ cells, TIM-3 marks a dysfunctional or exhausted phenotype and limits effector function [14].

TIM-3 ligands include cell surface proteins CAECAM1, phosphatidyl serine (PtdSer), soluble protein high mobility group box 1 protein (HMGB1), and galectin-9 (Gal-9), which can be expressed on the cell surface or can be secreted [14]. CEACAM1 is primarily expressed on cancer cells but can also be expressed on CD8+ and CD4+ T cells. PtdSer is expressed on apoptotic cells, and Gal-9 is expressed on cancer cells and APCs. HMGB1 is a soluble protein that binds to DNA from dying cells [14] (Figure 1).

CEACAM1 is expressed constitutively on epithelial cells and in a regulated manner on immune cells depending on their activation state. Defective CEACAM1 inhibits TIM-3-mediated inhibitory signaling, suggesting that TIM-3 is dependent on CEACAM1 for proper functioning [14]. Co-inhibition of TIM-3 and CEACAM1 results in improved elimination of CRC tumors in mice [74]; however, no clinical trials are currently investigating this combination for CRC. PtdSer is released when cells undergo apoptosis and directly regulates TIM-3 function [75]. HMGB1 binds DNA released from dying cells and mediates cytokine production and activation of an innate immune reaction that is inhibited upon TIM-3-HMGB1 binding [14]. Gal-9 is expressed on and can be secreted by tumor cells and APCs, and upon its binding to TIM-3 induces cell death in Th1 cells. Most anti-TIM-3 therapies target its interaction with CEACAM1 or PtdSer [76]. Recent investigation revealed that PD-1 physically interacts with Gal-9 and TIM-3 and attenuates Gal-9/TIM-3-induced apoptosis [77]. There is currently one clinical trial (NCT02817633) recruiting patients with advanced solid tumors for treatment with TSR-022, an anti-TIM-3 antibody [78]. Further investigation of anti-TIM-3 tumor suppressive mechanisms and its clinical utility is needed to optimize treatment combinations with other ITs. It is clear that while the driving mechanisms of different ICBs are similar, their cellular expression patterns differ which could have a profound effect on their roles in anti-tumor immunity and could translate to synergistic mechanisms when these therapies are combined.

#### 2.1.5. Anti-NKG2

The Natural Killer Group 2 (NKG2) family of receptors is expressed on a majority of NK cells and a subset of CD8+ T cells. Different NKG2 isoforms activate either immunosuppressive or immunostimulatory pathways. There are five isoforms, namely NKG2A, NKG2B, NKG2C, NKG2D, NKG2E, and NKG2F [15]. Inhibitory NKG2 receptors include NKG2A and NKG2B, while stimulatory receptors include NKG2C, NKG2D [16], and NKG2E. NKG2F has an unknown function. NKG2A/B bind to HLA-E, a ligand expressed at low levels with a wide tissue distribution [15,79]. Binding inhibits myriad things including Ca^2+^ influx, degranulation, cytokine production, and NK cell proliferation [16]. NKG2D is the most well-understood NK cell stimulatory receptor [80]. It binds to MHC class I chain-related proteins A and B (MICA and MICB), which are expressed at low levels on healthy tissue and are upregulated in the presence of DNA damage or cellular transformation. NKG2D cognate ligand binding results in enhancement of NK cell cytotoxic potential. Currently, there are no clinical trials targeting NKG2 inhibitory isoforms [81]; however, there are NKG2D CAR-T cell-based therapies under clinical evaluation including CYAD-101, NKR-2, and KD-025. All are in early recruitment phases [82,83,84,85,86,87]. Similar to other receptors relevant to ICB-based therapy, NKG2 has a wide variety of roles that may contribute to tumor immunity or tumor progression in unknown ways. The results of these trials are highly anticipated as the field continues to focus on improving responses to IT in CRC.

### 2.2. Adoptive Cell Therapies

Adoptive cell therapy, which is also known as adoptive cell transfer, cellular adoptive IT, or T cell transfer therapy, is a type of IT in which a patient’s T cells are removed from their blood, expanded ex vivo, sometimes modified to enhance their anti-tumor activity, and then administered back to the patient [53]. Adoptive cell therapies include tumor-infiltrating lymphocyte (TIL) therapy, engineered T cell receptor (TCR) T cell therapy, chimeric antigen receptor T cell (CAR-T cell) therapy, and NK cell therapy [53,88]. ACT is highly personalized to the patient; however, it is generally expensive and difficult to manufacture. Possible side effects include a sudden increase in cytokine levels that can result in mild fever, nausea, headache, rash, rapid heartbeat, low blood pressure, or trouble breathing. Graft versus host disease and B cell aplasia are other risks that come with ACT. A rare side effect of TIL therapy is capillary leak syndrome, which could result in dangerously low blood pressure [89]. Despite these limitations and toxicities, these personalized therapies confer immune memory due to permanent modification of anti-tumor immune cells [56].

#### 2.2.1. Tumor-Infiltrating Lymphocyte Therapy

TILs are lymphocytes that accumulate at the tumor margins or infiltrate within the tumor. These TILs are a heterogeneous population of cells that recognize tumor-associated antigens [TAAs) that are uniquely expressed or overexpressed by cancer cells. These cells can be isolated from single-cell suspensions from tumor samples collected during surgery. These tumor samples are homogenized and cultured using various protocols designed to massively expand the TILs, such as culturing them in the presence of specific cytokines or APCs, and are then reinfused into the patient [53]. It was demonstrated as early as 1986 that the adoptive cell transfer of TILs to mice bearing MC-38 colon adenocarcinoma tumors caused significant regression of liver and lung micrometastases [90]. There are currently no approved TIL therapies for CRC; however, numerous therapies are under evaluation in clinical trials [91].

#### 2.2.2. Engineered T Cell Therapy

As T cells play a key role in mediating anti-tumor immunity, efforts have been made to genetically engineer or modify T cells to enhance their anti-tumor effects. Therapies developed from these efforts include TCR T cell therapy and CAR-T cell therapy. The modification of T cells can involve viral vector transfection using a lentivirus or retrovirus to express specific TCRs or CARs on the T cell surface. Both of these therapies are autologous, meaning they involve extraction, ex vivo modification, and re-infusion of a patient’s own T cells. However, their antigen recognition mechanisms differ. TCRs use heterodimers to recognize MHC class I-presented peptides on tumor cells, while CAR-T cells use antibody fragments to directly bind to antigens on cancer cells (Figure 2). Because the peptides presented by TCRs have gone through processing by APCs, TCRs can recognize intracellular proteins, while the antigens that are recognized by CAR T cells must be expressed on the cancer cell surface [92]. Costimulatory molecules are often administered with engineered T cells, including CD28 or 4-1BB (CD137) [92]. Low efficacy due to a lack of tumor-specific antigens and adverse events after CAR-T treatment of patients with solid tumors necessitates new approaches including the addition of immune-activating molecules on CAR-T cells, regional administration, bispecific engagers, and optimization of CAR structure [93]. Very few clinical studies evaluating TCR T cells are underway, but one ongoing trial is investigating the use of TCRs that were cloned from neoepitope-targeting CD8+ cells in the patient’s blood for treatment of solid tumors [94].

#### 2.2.3. Natural Killer Cell Therapy

Natural killer (NK) cells are an essential component of the anti-tumor response. Similar to cytotoxic CD8+ T cells, NK cells recognize and directly kill cancer cells through the release of cytotoxic granules containing perforin and granzyme or through other pathways involving FasL/Fas. NK cells also produce IFN-γ and TNF-α as well as other important anti-cancer cytokines. Due to their evident role in eliminating cancerous cells and enhancing the anti-tumor response, NK cell-based therapies have been a topic of interest in cancer therapy development. These therapies involve autologous transfusion of NK cells that have been expanded, primed, and/or modified to enhance tumor cell killing. Similar to CAR-T cell therapy, CAR-NK cell therapy involves genetic modification of NK cells to enhance their recognition of cancer cell antigens [95].

CAR-NK cells in development for treatment of CRC include those that recognize epithelial cell adhesion molecule (EpCAM/CD326), which is overexpressed in many solid tumors and upregulates oncogene expression and cell proliferation [17]. NK cells have also been modified to recognize carcinoembryonic antigen (CEA) [95], which for many years has been the only widely recommended prognostic biomarker for CRC [96]. CEA levels are expected to fall after surgical resection of the tumor [97] and overexpression may contribute to human cancer progression by inhibiting cell differentiation, apoptosis, and anoikis [28]. Preclinical evaluations have thus far demonstrated positive anti-tumor activity.

Several NK cell therapies have been translated into clinical trials, one of which is indicated for MUC1-positive solid tumors including CRC [98]. MUC-1 is aberrantly overexpressed in CRC [36] and preliminary results of this trial show promise [95]. Other trials involve combination of NK cell-based therapies with various other treatments including radiotherapy [99], cetuximab [100], bevacizumab [101], and chemotherapy [102].

### 2.3. Monoclonal Antibodies

mAbs used for cancer therapy are a homogenous pool of antibodies that are specific for antigens which are overexpressed in cancer cells. Such antigens are often referred to as cancer-associated antigens (CAAs). mAbs may also be used for imaging when paired with a marker such as a radionuclide [103]. Early development of mAb therapy involved naked mAbs, which simply bind to a target and prevent its downstream signaling. More sophisticated approaches have been taken to improve efficacy of mAb therapy including the conjugation of mAbs to toxins or therapeutic agents, as well as the generation of bispecific mAbs which recognize distinct epitopes on each end of the molecule. In addition to blocking downstream signaling, IgG mAbs can induce antibody-dependent cell-mediated cytotoxicity (ADCC) via binding of the Fc receptor to immune cells [104]. Advantages of mAbs include their specificity, effectiveness across many cancer types, ease of conjugation, and low cost compared to other ITs. However, some limitations exist including an expensive target identification process, common inadequate pharmacokinetics and tissue accessibility, and development of resistance mechanisms [56].

#### 2.3.1. Naked Monoclonal Antibodies

The majority of mAbs that are used in clinical practice to treat CRC are naked mAbs. These include cetuximab (anti-EGFR), panitumumab (anti-EGFR), bevacizumab (anti-VEGF), nivolumab (anti-PD-1), pembrolizumab (anti-PD-1), and ipilimumab (anti-CTLA-4).

mAbs have been developed to target EGFR, a receptor that is often overactive in CRC cells and triggers cell proliferation. Cetuximab is a chimeric mAb which contains a human constant region and a murine variable region, and is FDA-approved for mCRC patients who overexpress EGFR and who are refractory to or intolerant of irinotecan-based chemotherapy. Panitumumab is a human mAb for mCRC patients who overexpress EGFR and who have failed fluoropyrimidine-, oxaliplatin-, and irinotecan-containing chemotherapy regimens. Panitumumab has a higher binding affinity for EGFR, and due to the absence of murine components, has a longer half-life and causes fewer infusion reactions compared to cetuximab [18]. The overall survival of patients receiving cetuximab (10 months) and panitumumab (10.4 months) was similar in the phase III ASPECCT study, indicating their comparable efficacy (HR 0.97; 95% CI 0.84–1.11) [2,105]. Anti-EGFR mAbs are thought to induce ADCC via binding of the Fc receptor to immune cells, though the extent of ADCC induction across EGFR inhibitors varies and the contribution of ADCC to the mechanism of cetuximab and panitumumab is questionable [2,104]. New anti-EGFR mAbs in development such as GA201 and imgatuzumab have been glycoengineered to enhance intrinsic ADCC induction [106,107]. A majority of CRC patients receiving anti-EGFR therapy experience dermatological toxicity in the form of a skin rash. 50% of patients receiving cetuximab for more than 6 months experience hypomagnesemia. Infusion reactions occur in 3.5–7.5% of patients [108].

VEGF is a soluble factor that stimulates the production of blood vessels, and is often overexpressed in CRC. Bevacizumab, which is FDA-approved for the treatment of mCRC [18], binds to soluble VEGF and prevents binding of this growth factor with its receptors VEGFR-1 and VEGFR-2. Both wild-type and mutated KRAS tumors, as well as both left- and right-sided tumors, seem to respond well to bevacizumab in both the first- and second-line settings [2]. Anti-EGFR and anti-VEGF mAbs have been directly compared in the clinic, revealing that cetuximab is likely more effective against left-sided, wild-type RAS tumors [2]. As a first-line therapy, cetuximab might improve the efficacy of subsequent bevacizumab therapy for these patients [109]. Side effects associated with bevacizumab, include hypertension, proteinuria, epistaxis, upper respiratory infection, anorexia, stomatitis, diarrhea or other gastrointestinal symptoms, headache, dyspnea, fatigue, and exfoliative dermatitis [110].

As discussed above, the ICB mAbs ipilimumab (+nivolumab) and pembrolizumab are approved for MSI-H mCRC tumors.

#### 2.3.2. Conjugated Monoclonal Antibodies

The feasibility and clinical efficacy of conjugated mAbs for treatment of CRC was demonstrated decades ago in a 1991 study of mCRC patients treated with either a CEA- or tumor-associated glycoprotein (TAG)-targeting mAb conjugated to iodine-13 [111]. Since then, several other studies have demonstrated that conjugated mAbs can be efficacious for treatment of CRC. Panitumumab has been conjugated to platinum agents and demonstrated excellent in vivo efficacy in reducing CRC tumor size [112]. Specific populations of cells such as stem cells, which express cell surface marker CD133, within the TME can be targeted with conjugated mAbs. Suppressed tumor growth and decreased recurrence was observed in an HCT116 xenograft model after treatment with anti-CD133 antibody-conjugated SN-38 nanoparticles [19]. HER3, a growth factor receptor that is overexpressed in CRC, has been targeted with U3-1402, a novel HER3-antibody conjugated to a topoisomerase I inhibitor. In vivo investigation of U3-1402 demonstrated tumor regression independent of KRAS status [20]. Following the success of the HER2-targeting mAb trastuzumab conjugated with the DNA topoisomerase I inhibitor deruxtecan (together known as ENHERTU) in the phase 2 DESTINY-Breast01 trial, this therapy was evaluated in several other types of solid tumors [113] including CRC (DESTINY-CRC01 [114]. Results of the DESTINY-CRC01 trial demonstrated durable responses in patients with previously treated HER2-positive mCRC (median progression-free survival 6.9 months, 95% CI = 4.1 months–not evaluable) [23], further supporting the use of conjugated mAbs in the targeted treatment of CRC in the adjuvant setting.

#### 2.3.3. Bispecific Antibodies

Bispecific antibodies (bsAbs) are antibodies that recognize two distinct epitopes on either end. Bispecific antibodies can be separated into combinatorial or obligate. Combinatorial bsAbs have a function or activity that could be achieved by combining two separate mAbs whereas obligate bsAbs have a function or activity that cannot be achieved with two separate mAbs, with the latter being a more attractive format due to the potential of creating novel functionality. Obligate bsAbs create novel functions by (1) bridging cell types such as cytotoxic CD8+ T cells and cancer cells to induce T cell-mediated killing, (2) crosslinking receptors on cells to either inactivate or activate them, (3) positioning an enzyme and substrate as a cofactor mimetic, or 4) using one binding end to gain access into a restricted cellular compartment and using the other to exert a therapeutic effect, often referred to as “hijacking” or “piggybacking” [24].

Most bsAbs are T cell-engaging bsAbs (bsTCEs) that engage T cells using CD3, a cell surface protein expressed on both CD8+ and CD4+ T cells. bsTCEs have shown great preclinical efficacy. For example, a bsTCE targeting GPA33, which is expressed on >95% of CRC tumors, was used to treat mice with colon or gastric cancer xenografts. Treatment resulted in tumor regression across MSS status [25]. bsTCEs have been developed to target guanylyl cyclase C (GUCY2C), which is also overexpressed in CRC. Treatment using GUCY2C-targeting bsTCEs demonstrated T cell–mediated killing of CRC cells in vitro and in vivo in CRC xenograft models across KRAS and BRAF status. Furthermore, combination of GUCY2C-targeting bsTCEs is enhanced when combined anti-PD-1/anti-PD-L1 treatment or anti-angiogenic therapies [27]. Treatment of CEA-expressing human CRC xenografts with a CEA/CD3-binding bsTCE induced an inflammatory response in the TME and caused tumor regression [115]. bsTCEs have also been designed to specifically target CD133+ stem cells in the TME and have shown efficacy in in vivo CRC models [116]. Development of resistance to bsTCEs may involve upregulation of checkpoint molecules PD-1/PD-L1; however, this may be overcome by combination treatment of bsTCEs with anti-PD-1 or anti-PD-L1 antibodies [24].

Mutation-associated neoantigens (MANAs) are proteins encoded by mutated cancer driver genes that have gone through proteolytic processing within the cell and are presented on the cell surface by human leukocyte antigen (HLA) molecules. TCR-mimicking antibodies directed toward these MANAs (MANAbodies) have been developed into bsTCEs that target mutant KRAS or TP53, two genetic alterations frequently found in CRC. These bsTCEs can recognize and kill cancer cells with very low levels of their cognate antigen in vitro and in vivo [117,118,119].

Gamma delta (γδ) T cells are unconventional T cells that play a major role in the T cell-mediated anti-tumor response despite the fact that they represent only a small fraction of the total T cell population in peripheral blood [120]. bsTCEs targeting EGFR-positive cancer cells for γδ T cell-mediated death have been developed and have demonstrated efficacy in killing CRC cells while sparing EGFR+ keratinocytes, suggesting that this treatment may have a high therapeutic index in patients [24].

Some bsAbs are not T cell-engaging but can suppress tumors by different mechanisms. For example, vanucizumab, a bsAb targeting VEGF-A and angiopoietin-2, demonstrated acceptable tolerability and decreased tumor vascularity in a heterogeneous population of patients with solid tumors (13/42 had CRC) [31]. In a CRC xenograft model, vanucizumab in combination with chemotherapy was superior compared to the clinical standard of anti-VEGF and chemotherapy combination treatment [121]. Additionally, bsAbs have been conjugated with toxic particles, inducing both photothermal and immune-mediated killing of cancer cells in a heterotopic colorectal tumor model [122].

A disappointing phase I study investigating LY3164530, a bsAb targeting MET and EGFR, demonstrated the important limitation of bsAb fixed stoichiometry. Final results indicated limited efficacy and significant toxicity due to EGFR inhibition. EGFR inhibition causes more severe toxicity than MET inhibition, but because stoichiometry of bsAbs is fixed, it is difficult to adjust relative affinities on either end of the molecule. This indicates that targets with vastly different therapeutic indices are likely not an ideal combination for the bsAb approach [32]. Several other clinical trials are underway to investigate both obligate and combinatorial bsAbs (Table 3), for example XmAb841, a bsAb against CTLA-4 and LAG-3 either alone or in combination with pembrolizumab [123].

#### 2.3.4. Nanobodies

Some limitations of mAbs for the treatment of cancer include inadequate pharmacokinetics and tissue accessibility [124]. The emerging field of nanobodies (Nbs) may solve some limitations of mAbs and their derivatives, as these single-variable domains of the camelid antibody are smaller, more stable, and have better tissue penetration compared to traditional mAbs [125]. One clinical trial involved treatment of four patients with solid tumors (1 of which was a CRC patient) with TAS266, a nanobody that targets death receptor 5 (DR5), a cell surface receptor that triggers apoptosis [34]. However, the trial was terminated early due to dose-limiting toxicity in three patients [125,126]. Another interesting early phase I trial is recruiting mesothelin+ CRC and ovarian cancer patients for treatment with mesothelin-targeting CAR-T cells that secrete PD-1 Nbs [127].

### 2.4. Oncolytic Virus Therapy

Oncolytic viruses are viruses that selectively infect and kill or incapacitate cancer cells without harming normal cells. This selectivity is achieved through genetic engineering. There has been relatively limited clinical investigation of oncolytic viruses for treatment of CRC and no such therapies are FDA-approved [128]. One phase I/II trial evaluated a genetically engineered oncolytic herpes simplex virus (NV1020) in patients with previously treated mCRC. Researchers observed that NV1020 stabilized liver metastases and patients experienced minimal toxicity, justifying a phase II/III trial; however, this has not been initiated yet [129]. JX-594 is a vaccinia poxvirus with engineered loss of thymidine kinase, enabling its replication only in cells high levels of thymidine kinase [130]. JX-594 also contains an added GM-CSF gene, encoding a cytokine that induces DC differentiation, maturation, and function, and increases T cell activity. JX-594 did not cause dose-limiting toxicity and achieved stable disease in 67% (*n* = 10) of pre-treated CRC patients [5,131,132]. Combination of JX-594 with the anti-PD-L1 antibody durvalumab demonstrated tolerability and potential efficacy in mCRC patients (median overall survival 7.5 months, CI 4.9–10.1) [133]. The Newcastle disease virus (NDV) is an interesting natural RNA oncolytic virus. The NDV envelope protein and other intracellular factors induce an immune response to elicit a potent anti-tumor effect in a variety of cancers including CRC. Genetic modification of NDV could enhance this tumor lysis activity or could allow NDV to act as a vector for delivery of therapeutic genes [134]. Advantages of oncolytic virus therapy include specificity, and that it may induce an “immune priming” mechanism that could be utilized to improve response to other ITs [135]. One limitation of oncolytic viruses is that the host anti-viral immune response may reduce efficacy [56].

### 2.5. Vaccines

Cancer vaccines, or tumor vaccines, are a type of treatment that helps the body’s immune system recognize and destroy cancer cells. Treatment generally involves direct administration of cells or antigens to the patient (whole-tumor or peptide vaccines, respectively), indirect antigen delivery (viral vector), or delivery of APCs that were trained ex vivo to recognize cancer cells. Following antigen delivery, APCs process and present the peptide to CD8+ T cells, stimulating anti-tumor immunity. Adjuvants, or substances designed to enhance or direct the immune system toward a cytotoxic response, are often given in combination with the vaccine. Cancer vaccines are generally therapeutic, meaning that they are administered after the cancer has been diagnosed, as opposed to a preventative vaccine that is administered pre-diagnosis. Advantages of cancer vaccines include their specificity and personalization; however, the delivery of foreign antigens through this treatment could induce rejection [56].

#### 2.5.1. Whole Tumor

Whole-tumor, or whole-tumor-cell, vaccines involve direct administration of whole tumor cells to a patient. These tumor cells can either be autologous or allogenic. In theory, whole-tumor vaccines train the immune system to recognize the entire repertoire of antigens that are present in the heterogeneous TME. Intriguingly, in preclinical CRC models, whole-tumor autologous vaccines exacerbated tumor growth when delivered therapeutically [136]. Initial clinical trials with autologous whole-tumor vaccines in CRC patients were similarly unimpressive [137,138,139]. Besides inefficacy, limitations of autologous vaccines include the large number of tumor cells needed, restricting this therapy to more advanced-stage patients with resectable tumors [140]. In contrast to autologous vaccines, allogenic vaccines use cell lines instead of tumor cells and have demonstrated improved efficacy in patients with colon cancer [136,140].

#### 2.5.2. Peptide

Peptide vaccines involve administration of immunogenic cancer-associated antigens to the patient. Several CRC-associated antigens have been targeted with peptide vaccines preclinically, including ephrin type-A receptor 2 (EphA2), CEA, MUC-1, Survivin, and SART3. Though clinical evaluation has revealed that peptide vaccines can cause strong anti-tumor immune responses in CRC patients, no correlation with improved clinical outcome has been made thus far [40,128].

#### 2.5.3. Viral Vector

Viral vector vaccines use a viral vector system to deliver TAAs. Some studies suggest that vector vaccines tend to generate a higher immune response compared to peptide vaccines, but clinical trials have revealed no difference between treatment groups in terms of immune response or clinical outcomes in CRC patients [128].

#### 2.5.4. Dendritic Cell Therapy

The most promising type of cancer vaccine involves vaccination with DCs, in which DCs are removed from the patient, expanded and modified ex vivo to recognize TAAs, and then re-infused back into the patient to stimulate the cytotoxic T cell response [128]. The most commonly known DC therapy is Sipuleucel-T for treatment of prostate cancer, which improves survival of patients by approximately 4 months [141]. For CRC, however, results have not been comparable. A randomized phase II clinical trial utilizing an autologous tumor lysate DC vaccine induced a tumor-specific immune response in patients, but did not impact on survival compared to patients receiving the standard of care [128].

### 2.6. Immune System Modulators

Immune system modulators are proteins or compounds that induce or direct a tumor-suppressive immune reaction. These include cytokines such as interleukins and interferons as well as compounds such as thalidomide. Cytokines and chemokines are fundamental to immune responses including the anti-tumor immune response. On a basic level, they regulate the growth of all blood cells and aid the development of immune responses in the event of bacterial infection, viral infection, or cancer development. In recent decades, researchers have begun to uncover complex roles of cytokines that contribute to both pro- and anti-tumor effects that are sometimes disparate across cancer types [142,143,144]. Context-dependent effects of cytokines should be taken into consideration in the development of immunomodulatory therapies, making their development somewhat challenging. It is also likely that molecular features such as high PD-L1, which are often found in colorectal tumors, will counteract the anti-tumor mechanisms of immune system modulators, necessitating combination therapies. Another limitation of immune system modulators is their low specificity and risk of immediate onset cytokine response syndrome, also known as cytokine storm. Nonetheless, immune system modulators have several advantages including the fact that many are already FDA-approved, their small size which facilitates access to cancer cells, their relatively low cost, and their ability to generate a general anti-cancer immune response that may oppose development of resistance [56]. Several immune system modulators have entered clinical phases of investigation (Table 3).

#### 2.6.1. Interleukins

Interleukins (ILs) are a class of cytokine that are produced primarily by leukocytes. There are currently 38 recognized interleukins with some containing multiple family members such as IL-1α and IL-1β for an overall total of more than 60 [145]. Interleukins can have a wide variety of pro- or anti- tumor effects and many have been therapeutically targeted in CRC clinical trials including IL-2, IL-12, IL-11, IL-6, IL-α, and IL-1β. IL-2 and IL-12 are potent activators of NK cells and CD8+ T cells. IL-11 and IL-6 are tumor-promoting cytokines that function by various mechanisms [41]. It is well established that IL-1α and IL-1β contribute to pro-tumor and anti-tumor mechanisms and have pleiotropic effects on immune cells, angiogenesis, cancer cell proliferation, migration, and metastasis [41,44,45], and therefore there is much controversy over whether or not they should be inhibited in cancer. Treatment of mCRC patients via activation IL-2 or IL-12, or inhibition of IL-11, IL-6, IL-1α, or IL-1β, is currently under clinical investigation. Most are in combination with several other treatment regiments and evaluate efficacy enhancement of these other therapies upon addition of IL. Other ILs have been investigated preclinically, yielding results that may support clinical translation. For example, IL-33 has been shown to have potent anti-tumor activity against CRC cells that is dependent on the presence of eosinophils, which are recruited, activated, and degranulated by IL-33 [46].

#### 2.6.2. Interferons

Interferons (IFNs) make up another a class of leukocyte-produced cytokines. There are three types of IFN: Type I, II, and III. Type I IFNs include 17 distinct proteins that fall into subtypes IFNα, IFNβ, IFNε, IFNκ and IFNω, but only IFNα and IFNβ have been well studied in the context of cancer. There is only one Type II IFN, IFN-γ, and Type III IFNs include IFNλ1, IFNλ2 and IFNλ3. Type I and Type III IFNs are induced by pattern recognition receptor pathways, while Type II IFN-γ is mainly induced by certain types of mitogens or cytokines including IL-12 and IL-18 produced by T cells and NK cells [146]. The role of IFNs in cancer has largely been described as anti-tumor, though evidence suggests that in certain contexts IFNs such as IFN-γ can be pro-tumor at certain doses [147]. As mentioned above, context should be carefully considered when developing therapeutic IFN treatments.

The role of IFN dysregulation in CRC is evidenced by the fact that single-nucleotide polymorphisms (SNPs) in the IFN I and IFN II pathways impact risk of CRC development and survival post-diagnosis [146,148]. Thus, IFNs have been developed as a target for CRC therapy. IFN-γ is likely the most well-studied IFN for treatment of CRC, but its role in cancer is not clear [149,150]. IFN-γ cross-talk with immunostimulatory M1 macrophages can inhibit tumor growth. Stimulation of IFN-γ production in T regs in combination with anti-PD-1 ICB caused tumor regression in MC-38 tumor-bearing mice that were resistant to single treatment with anti-PD-1. Interestingly, because the IFN-γ receptor is expressed at higher levels on CRC stem-like cells compared to other CRC cells, IFN-γ treatment can selectively induce apoptosis in these cells [147].

Other preclinical studies have shown less promising results. For example, mice with CT26 colon cancer tumors exposed to IFN-γ experienced downregulation of antigen presentation and resultant immune evasion [150]. Another study demonstrated that the tumor-suppressor gene SOCS1 mediates its anti-oncogene function through negative regulation of IFN-γ [150,151]. Not surprisingly, clinical trial results have been disappointing or inconclusive. A 1995 trial suggested inefficacy of surgical adjuvant treatment for patients with high-risk colon cancer [152]. Another study combined IFN-γ with 5-FU and anti-VEGF therapy bevacizumab, but the specific contribution of IFN-γ to the treatment effect was not investigated [153,154]. There is an ongoing trial that involves treatment with IFN-γ and the anti-PD-1 therapy nivolumab for treatment of solid cancers including MSI-H CRC. Preliminary results from other initial clinical trials indicated that low or moderate doses of IFN-γ were more effective than high doses [150], suggesting that dose should be carefully considered when administering treatment and interpreting study results.

#### 2.6.3. cGAS-STING

The cyclic GMP-AMP synthase (cGAS) stimulator of interferon genes (STING) pathway is the primary sensor of cytosolic double stranded DNA (dsDNA) from either invading pathogens or the cell itself when it becomes cancerous or begins to die. In response to cytosolic DNA, the cGAS-STING pathway induces Type I IFNs, other inflammatory genes, and cellular senescence to inhibit the development of cancer [50]. STING expression in CRC has independent prognostic value [155] and a STING-related prognostic score may allow identification of high-risk CRC cases [156]. The STING agonist E7766 has shown promise in the preclinical setting in terms of anti-tumor activity and induction of a tumor-specific memory response in a murine metastatic liver CRC model. E7766 is currently in clinical trials for treatment of advanced solid tumors including CRC [157].

#### 2.6.4. Immunomodulators

Immunomodulatory drugs (IMiDs) therapeutically modulate the immune response to provide patient benefit by various mechanisms. In the context of cancer, IMiDs usually refer to thalidomide and its analogues lenalidomide and pomalidomide. Thalidomide inhibits angiogenesis, enhances production of TNF, and stimulates NK cells. Following the discovery of its anti-angiogenic properties, thalidomide was administered to drug-resistant multiple myeloma patients who subsequently experienced tumor regression [158]. Subsequent investigation of thalidomide, lenalidomide, and pomalidomide treatment in other cancer types in vitro and in vivo indicated that IMiDs bias the immune response toward anti-tumor immunity, inhibit metastasis, and alter cancer cell signaling [159]. To date, two clinical trials involving thalidomide treatment have been completed. In one of these trials, thalidomide and capecitabine were administered to mCRC patients and the rate and duration of disease stabilization suggested some efficacy [160].

### 2.7. Targeting the Immunosuppressive TME

Tumors with immunosuppressive TMEs are characterized by presence of certain cell types such as myeloid-derived suppressor cells (MDSCs), M2-like tumor-associated macrophages (TAMs), and T regs, and lack of CD8+ T cells. These tumors are often classified as “cold,” whereas tumors with large numbers of infiltrating, active CD8+ T cells are classified as “hot.” One strategy to transform a cold tumor into a hot tumor involves inhibition of immunosuppressive cell types. One clinical trial is investigating the CCR5 inhibitor Maraviroc for treatment of mCRC. CCR5 is receptor that is expressed on CRC cells, MDSCs, and T regs, thus its inhibition can both slow the cancer cell growth and inhibit immunosuppression in the TME [51,52].

## 3. Predictive Biomarkers for Response to IT in Colorectal Cancer

Similarly to traditional cytotoxic therapies, IT for treatment of CRC comes with risks of side effects such as autoimmunity or cytokine storm, in which the concentration of cytokines in the blood is high enough to cause harmful inflammation and immune effects [161]. As with any therapy that causes side effects, development of predictive biomarkers is a crucial step in preventing toxicity by identifying specific patient populations who are likely or unlikely to benefit from the therapy.

Currently in CRC, physicians may predict response to IT based on the genetic alterations that are present in the patient’s tumor, specific characteristics of the CRC TME, and specific serum-based (liquid) biomarkers present in patient serum. Some biomarkers used to predict response to IT in CRC are based several key molecular characteristics of this disease, such as specific genetic alterations. Others have been developed based the mechanisms of IT, including neoantigen load, antigen presentation efficiency, diversity of the TCR repertoire, activity and abundance of immune cells, expression of certain receptors on immune cells or cancer cells, and/or levels of pro- and anti-tumor cytokines. Additional biomarkers have been developed that may be used to predict response to other types of anti-cancer therapy such as levels of ctDNA and presence of intratumoral microbes or circulating microbe-associated compounds. These biomarkers are summarized in Figure 3 and are explained in more detail in the following subsections. In general, predictive biomarker development has mostly focused on ICB, though each may also have predictive value for various other ITs if their overarching mechanisms are similar.

### 3.1. Prediction Based on Genetic Alterations

Three genetic alterations that are commonly found in CRC include chromosomal instability, MSI, and CpG island methylation [162]. Presence of each alteration can indicate how likely CRC patients are to respond to IT.

#### 3.1.1. Chromosomal Instability

The majority of CRC cases exhibit a chromosomal instability (CIN) phenotype characterized by a progression of genetic alterations and their resultant histological changes. Common and predicable genomic alterations include the activation of proto-oncogenes *KRAS* and *BRAF* and the inactivation of the tumor-suppressor genes *APC* and *TP53*. It is clear that CIN is associated with poor prognosis in multiple tumor types including CRC; however, it is currently unclear if CIN is a biomarker of response to IT [163,164]. Mutations in oncogenes such as *KRAS* and *BRAF* have recently been shown to impact on immunomodulation [165]. For example, *KRAS*-mutant patients have more T regs and less activated CD4+ memory T cells as compared to *KRAS* wild-type patients, resulting in a more immunosuppressive TME [166]. Moreover, there are several reports linking *KRAS* mutations with enhanced PD-L1 expression on tumor cells, which results in reduced T cell functionality. However, there are limited data on the role of these mutations in the prediction of response to IT [167].

The inactivation of tumor-suppressor genes can also play a role in CRC response to IT. For example, the *APC* tumor-suppressor gene is involved in the cellular transition from G1 to S phase and loss of *APC* can result in the sustained activation of the Wnt signaling pathway and, thus, increased nuclear β-catenin [168]. It has been shown that both MSI-H and MSS CRC patients with *APC* biallelic mutations have decreased T cell tumor infiltration [169]. Another commonly mutated tumor-suppressor gene in CRC is *TP53*. The encoded protein p53 plays a role in controlling cell cycle and apoptosis. While p53 mutations in CRC classically confer resistance to conventional chemotherapies, they provide an additional benefit in the context of IT, as p53 mutated tumors often harbor an increased number of neoantigens. In contrast, wild-type p53 has been shown to enhance MHC class I peptide transport via upregulation of TAP1, the transporter associated with antigen processing [170]. Therefore, the absence of wild-type p53 in cancer cells can diminish cytotoxic T cell-mediated tumor cell death that may impact on IT outcomes in patients.

#### 3.1.2. Microsatellite Instability

Approximately 15–20% of cases of CRC exhibit the MSI-H phenotype, which is characterized by mutations in the MMR system [4]. Patients with MSI-H tumors have dMMR systems and thus, an accumulation of errors in microsatellite regions. In contrast, those MSS tumors tend to have pMMR systems, and thus a lower number of DNA replication errors. dMMR and the resulting MSI-H phenotype have long been considered predictors of efficacy for ICB in multiple tumor types. In CRC, MSI-H status is predictive of long-term responses to ICB [171].

#### 3.1.3. CpG Island Methylation

Another common occurrence in CRC is the aberrant epigenetic regulation of gene expression. Hypermethylation occurs when methyl groups are covalently bound in regions of CG dinucleotides or CpG-rich areas of DNA in promoter regions. In instances of normal gene expression, CpG regions are normally maintained in the unmethylated state. Methylation of these promoter regions can lead to gene silencing in tumor-suppressor genes, which is denoted as CpG island methylator phenotype (CIMP) [162]. The presence of TILs is associated with a high degree of CIMP, along with presence of other mutations [172] which could have a positive effect on response to ITs that are dependent on T cell infiltration, such as ICB and bsTCEs. CpG island methylator phenotype-high CRC is also associated with PD-L1 status of primary tumors [173].

#### 3.1.4. Tumor Mutational Burden and Neoantigen Load

In 2020, the FDA approved pembrolizumab for high tumor mutational burden (TMB-H) solid tumors based on the results of the KEYNOTE-158 study [174]. TMB is a measurement of the number of mutations per megabase of DNA in tumor cells. Typically, TMB-H CRC tumors have more immunogenic neoantigens and a higher neoantigen load. Neoantigen load is a quantification of mutations that can be targeted by T cells and is a biomarker of response to IT. Neoantigen load is correlated with the presence of TILs in both MSI-H and MSS CRC tumors [55].

#### 3.1.5. Other Specific Genetic Alterations

Another frequently mutated signaling pathway in CRC is the PI3K/AKT pathway. Activating mutations in *PIK3CA* occur in 25% of CRC cases [175]. *PIK3CA* mutations are associated with CD8+ T cell infiltration, PD-L1 expression, and improved IT benefit in patients with MSS CRC [176]. The PI3K/AKT signaling pathway is negatively regulated by PTEN, a phosphatase that is mutated in approximately 18% of MSI-H CRC cases [177]. CRC studies have shown that a loss of *PTEN* results in increased PD-L1 expression [178]. Moreover, loss of *PTEN* has been reported to result in decreased TIL presence and an immunosuppressive TME [179]. Another common genetic alteration in CRC is mutations in TGFβ, a collection of growth factors that regulate myriad cellular processes through their interaction with TGFβ receptors [162]. TGFβ activation is predictive of poor prognosis in CRC and is predictive of ICB resistance in several tumor types [180,181]. The most common mechanism of activation of TGFβ signaling in CRC is due to mutations in transforming growth factor β receptor type 2 (*TGFβR2*), which occurs in approximately 30% of CRC cases [182]. Mutated in *TGFβR2* is predictive of resistance to ICB in non-small-cell lung cancer (NSCLC), but there are currently limited reports concerning the prognostic significance of *TGFβR2* mutations in the context of CRC [183].

### 3.2. Prediction Based on the Tumor Microenvironment

The CRC TME is comprised of tumor cells, tumor-associated cells, endothelial cells, immune cells, and the extracellular matrix, all of which can be immunostimulatory or immunosuppressive and can influence the response to IT.

#### 3.2.1. PD-L1 Expression

PD-L1 is a tumor cell-expressed surface protein that is a common biomarker of response to IT across many tumor types. Tumor cells commonly overexpress PD-L1, and this overexpression is associated with poor overall survival across multiple tumor types [184]. Paradoxically, a study by Droeser et al. reported that PD-L1 expression is correlated with CD8+ T cell infiltration in MSS CRC tumors [185]. Ultimately, the value of PD-L1 alone as a predictive biomarker has limitations.

#### 3.2.2. Tumor-Infiltrating Lymphocytes

The presence and function of TILs such as T cells, B cells, and NK cells can be predictive of response to IT. A positive correlation between TILs and response to ICB has been shown repeatedly in melanoma [186]. A study by Li et al. reported that a signature of high immune cell infiltration in CRC patients was correlated with increased PD-L1 expression and better prognosis as compared to CRC patients clustered in the low and medium immune infiltration groups [187]. Interestingly, in CRC, a correlation between T reg cell tumor infiltration and response to chemo-IT has also been reported [188].

#### 3.2.3. Immune Status of the Tumor Microenvironment

The immune status of the TME can be broadly categorized as immune-inflamed, immune-excluded, or immune-desert. A study by Tang et al. categorized CRC patients into four groups based on the expression signatures of cancer-associated fibroblasts, MDSCs, M2-like TAMs, CD8+ T cells, and PD-L1 expression and found that classification of CRC patients into immune subtypes was a reliable predictor of prognosis. The clustering strategy developed here has great potential to guide personalized immunotherapy based on the TME [189].

#### 3.2.4. Diversity of T Cell Repertoires in TME

T cell receptors recognize specific antigens presented by MHC class I (for CD8+ T cells) or MHC class II (for CD4+ T cells) molecules. A process called genetic recombination occurs in T cells to rearrange the DNA at three loci (TRBV, TRBD and TRBJ) to develop TCRs that are specific for certain antigens. The T cell repertoire refers to all of the unique TCR genetic rearrangements within the adaptive immune system, thus the TME T cell repertoire refers to all of the unique TCR genetic rearrangements within the TME. Not surprisingly, having a diverse TME T cell repertoire is associated with better outcomes in response to immunotherapy. This holds true when evaluating CRC patient populations specifically. In a TCR repertoire analysis of advanced CRC patients treated with a combination of five cancer peptide vaccines and oxaliplatin-based chemotherapy, high TCR diversity scores were associated with improved response [190]. A recent report suggested a significant difference in the usage of TRBV and TRBJ genes between CRC patients and healthy controls, supporting its use as an additional TCR-based predictive biomarker in CRC [191].

#### 3.2.5. Tumor-Associated Macrophages

TAMs are a type of macrophage that create an immunosuppressive TME by producing immunosuppressive cytokines and growth factors, and by increasing immune checkpoint expression on immune cells. Macrophages are classified into two types: classical-activated M1 and alternate-activated M2. The conversion between these two states is called polarization. Though immunosuppressive TAMs can exhibit either polarization phenotype, it is believed that they generally are more M2-like [192]. Studies suggest that up to 80% of cancer patients, increased levels of TAMs predict worse outcomes. In CRC, however, there are conflicting findings as far as the prognostic and predictive value of TAMs [192,193,194] that may be clarified with investigations that make a clear distinction between macrophages with different polarization states or phenotypes. Our knowledge of the role of TAMs in the response to IT across cancer types is very limited [195]. In CRC, TAMs express PD-1 and treatment with anti-PD-1 compounds increased phagocytosis of PD-1+ TAMs. TAMs can also sequester anti-PD-1 antibodies, preventing their therapeutic binding to their target immune cell. Thus, TAMs play key roles in modulating response to immunotherapy in CRC cells but it is not clear whether their levels actually correlate with better or worse response to IT in CRC patients [196].

#### 3.2.6. The Gut Microbiota

The makeup of the gut microbiome may influence responses to immunotherapy. A recent investigation using a mouse model of CRC identified intratumoral presence of the bacterial species *Bifidobacterium pseudolongum*, *Lactobacillus johnsonii*, and *Olsenella* as enhancers of ICB efficacy. *B. pseudolongum* produced inosine, which translocated systemically after ICB-induced gut barrier impairment to activate anti-tumor T cells [197]. Another study found a significantly different ratio of *Prevotella/Bacteroides* species between ICB responders versus nonresponders in patients with GI cancer (*n* = 74; *p* = 0.032) [198]. Thus, the relative abundance of these species within the tumor could predict response to ICB but this remains to be confirmed by ongoing clinical trials [135].

### 3.3. Liquid Biomarkers

Peripheral blood offers a non-invasive source of potential biomarkers via liquid biopsies. Liquid biopsies can provide information on circulating factors such as peripheral blood cells, tumor DNA, and cytokines. Other potential liquid biomarkers that may predict responses in CRC include exosomes.

#### 3.3.1. Peripheral Blood Cells

There are significant differences in the proportions of lymphocytes, white blood cells, neutrophils and MDSCs between healthy controls and patients with CRC [199]. This differential peripheral blood cellular composition has been reported as both diagnostic and prognostic in patients with CRC. Peripheral NK cells are an independent predictor of survival in patients with CRC and their prognostic value increases when combined with B cell count [200]. In a study of CRC patients with liver metastases, patients more likely to respond to CAR T therapy had lower-fold changes in their neutrophil-to-lymphocyte (NLR) ratio [201].

#### 3.3.2. Circulating Tumor DNA

Circulating tumor DNA (ctDNA) is tumor-derived fragmented DNA found in the bloodstream that is no longer cell-associated. In a review of ninety-two clinical studies, ctDNA was demonstrated to be a reliable measure of tumor burden and response to therapy in CRC [202]. In a study of MSS CRC patients treated with combination therapy of regorafenib and PD-1 inhibitors, the authors found that ctDNA may be predictive of early therapeutic efficacy of IT in the MSS CRC patient population. Specifically, 10 patients with rising ctDNA levels or emergence of new clones 4 weeks after treatment experienced progressive disease after 2 months, whereas 3 patients with declining ctDNA experienced stable disease [65].

#### 3.3.3. Cytokines

As previously mentioned, cytokines play a key role in both pro- and anti-tumor immune responses. Predictive cytokines can be secreted by either the tumor cells or immune cells. Candidate cytokine biomarkers for CRC include APRIL, BAFF, IL8 and MMP2, which are inversely correlated with immune cell infiltration and expression of CD163, a marker of M2 macrophages [203]. Another investigation revealed that IL-17A increased expression of PD-L1 on CRC cells and that inhibition of IL-17A improved the efficacy of anti-PD-1 therapy in a murine MSS CRC model [204,205]. Further investigation and validation is needed to translate use of these biomarkers into the clinic.

#### 3.3.4. Exosomes

Exosomes are a membrane-bound type of extracellular vesicle, which can carry cargo such as mRNA, microRNA (miRNA), long non-coding RNA (lncRNA), protein, DNA fragments, or debris [206] and deliver this cargo to other cells, mediating various downstream signaling pathways. Extracellular vesicles contribute to tumorigenesis in many ways [206] such as by increasing the migration of fibroblasts in the TME [207]. Exosomes containing biomarkers such as PD-1 or PD-L1 predict better response to IT in melanoma and NSCLC, whereas exosomes containing hsa-miR-320b/c/d or hsa-miR-125b-5 predict worse response to IT in NSCLC. It remains to be seen if circulating exosomes predict response to IT in CRC [208].

### 3.4. Other Factors Influencing Response to IT

Factors other than molecular features of tumor and immune cells can influence patient response to IT. Sociological factors, lifestyle, and metabolic disorders such as obesity can have a profound impact as well, though this has only been minimally investigated in CRC. In other cancer types, exercise and a healthy diet promoted therapeutic benefit from ICB. Alcohol consumption decreases response rates and is associated with decreased mutational burden and antigen presentation by DCs [209,210,211]. Other interesting observations have been made, such as that obesity [212,213,214,215,216], smoking [217,218,219,220,221], and estrogen signaling [222,223] all contribute to higher PD-1/PD-L1 expression and thus better response to ICB in some cancers. There are little data on the relevance of these mechanisms in CRC response to IT, and it is a requisite topic of future investigation that could have great impact on the personalization of IT in the clinic [224]. One study suggested that CRC subtypes with low-level lymphocytic reactions may be vulnerable to the immunosuppression associated with smoking [225], and another study indicated that moderate intake of alcohol was inversely correlated with distal CRC [226]. These cancer-specific effects further necessitate investigation of these factors specifically in CRC.

## 4. Sensitizing CRC to IT

Development of IT for treatment of CRC has made great strides in recent decades, particularly for patients with MSI-H tumors. ICB targeting PD-1 and CTLA-4 have been approved and are often used for this subset of CRC patients in combination or with or after failure of conventional chemotherapy regimens. Despite this, a majority of CRC patients (85%) have MSS tumors that do not respond well to ICB. In this section, we describe the efforts that have been made to improve response rates.

Further investigation of the transcriptomic, cytokine, and immune cell responses to IT in MSI-H versus MSS CRC are likely to reveal differences that will guide combination treatment for MSS CRC patients. For example, the extent of immune cell, especially T cell, infiltrate is much higher in MSI-H CRC tumors and may correlate with positive response to IT [227]. Enhancing immune cell infiltration in MSS tumors, perhaps by inhibiting MDSCs, may be a promising strategy to sensitize these tumors to IT.

It is currently unclear if FDA-approved ICB treatments are more effective in the neoadjuvant or adjuvant setting, or if IT should be combined with chemotherapy or targeted treatments. Improving responses to IT in CRC will involve investigating the role of treatment timing and drug combinations. An important study revealed that neoadjuvant ipilimumab combined with nivolumab resulted in a 27% response rate in MSS tumors after only ~4 weeks of treatment (95% exact CI: 8–55%) [70], which is striking when compared to the low response rate reported by some clinical trials [2]. These results suggest that IT treatment in the neoadjuvant setting should be strongly considered for CRC patients.

One previously discussed study suggested that DCs are the major mediator of response to IT in CRC patients with MSI-H tumors. This finding supports the notion that the tumor microenvironment has profound effects on efficacy of IT and suggests co-administration of ICB with FLT3L, a DC-promoting cytokine [67,68].

Other exciting new developments could benefit both MSS and MSI-H CRC patients. Mucosal-associated invariant T (MAIT) cells are unconventional T cells that play a key role in the immune response to microbes. These cells have been most extensively studied in the context of CRC as compared to other cancer types, but still there is no consensus regarding the role they play in anti-tumor immunity. As MAIT cells are abundant in mucosal tissue such as the gut and are found in a majority of CRC tumors, elucidating this role may guide novel immunotherapies for CRC. Both agonist and antagonist molecules of the semi-invariant TCR on invariant T cells are well characterized and would ease the development of these therapies [228].

Another recent investigation determined that mutations in more than one hundred tumor-suppressor genes can prevent immune cell recognition of cancer cells in vivo. This novel insight into the role of altered tumor suppressors in immune evasion could guide novel immunotherapies which target or restore altered tumor-suppressor genes in combination with other IT [229,230].

Other work has identified cyclooxygenase (COX-2), the enzyme that produces the pro-survival and immune evasion protein prostaglandin E2 (PGE2) [231], as a promoter of colorectal tumorigenesis and MDSC immunosuppressive activity [232]. COX-2/PGE signaling is also thought to induce M2 macrophages while inhibiting T cell infiltration and NK cell/DC crosstalk [231]. Thus, inhibition of COX-2 may improve response to IT. Clinical trials combining PD-1 inhibitors with the COX-2 inhibitor celecoxib as neoadjuvant therapy are recruiting [231]. One trial (NCT03026140) will investigate this combination in both MSI-H and MSS CRC patients [233].

Ongoing investigation of immune escape mechanisms will continue to reveal next-generation IT targets. For example, investigation of the interplay of gut microbiota and IT may reveal approaches to increase abundance of immunostimulatory microbes and decrease abundance of immunosuppressive species to enhance IT efficacy [197,198,234].

Recent work has revealed that cancer cells hijack the mitochondria of immune cells via nanotubes. Inhibiting nanotube formation led to improved outcomes after anti-PD-1 therapy in a breast cancer model, and next steps should include similar investigation in colorectal cancer [235].

Other strategies involve identifying specific groups of patients within MSS CRC that are more likely to respond to treatment. One recent investigation revealed that patients with liver metastases had a lower response rate compared to patients without metastases (objective response rate 0% versus 19.5%, respectively), suggesting that IT should be more seriously considered for early-stage patients [236].

## 5. Discussion and Open Questions

Significant advancements have been made in recent decades as far as development of IT for CRC patients. The FDA approval of anti-PD-1 mAbs pembrolizumab and nivolumab, and anti-CTLA-4 mAb ipilimumab has improved the lives of many CRC patients and indicates the significant anti-cancer potential of IT. However, there are numerous topics of requisite future investigation to improve clinical response rates.

It is currently unclear if FDA-approved ICB treatments are more effective in the neoadjuvant or adjuvant setting, or if IT should be combined with chemotherapy. Recent studies have revealed a possible benefit of treating patients with first-line ICB, and other trials are investigating treatment combinations. It is likely that a majority of cancer patients will receive chemotherapy before or concurrently with immunotherapy in the near future. It is well documented that conventional chemotherapy and targeted therapies have drastic effects on cytokine production and secretion across cell types including cancer cells and immune cells. These cytokines can have effects that enhance the anti-tumor activity of the drug, but could also lead to acquired resistance of cancer cells or systemic toxicity in the patient. Cytokine profiling in cancer cells treated with these drugs can guide logical combination with IT to enhance their anti-tumor effects. This has been completed in HCT-116 CRC cells [237] and MCF7 breast cancer cells [238] treated with conventional chemotherapies and in HCT-116, HT-29, and KM12C CRC cells treated with MEK, TRK, tyrosine kinase, RET, BRAF, PARP, PI3K, and GSK-3 inhibitors as well as Imipridones [239]. In HCT-116 cells treated with chemotherapies 5-FU, irinotecan, oxaliplatin, cisplatin, or clinically relevant combinations, drug-specific effects and an impact of p53 status on the modulation cytokines IL-8, ferritin, and soluble TRAIL-R2 was observed [237]. Possible tissue-specific effects were noted after treatment of breast cancer cells and CRC cells with cisplatin, oxaliplatin, and 5-FU, including enhanced downregulation IFN-β and TRAIL and upregulation of TRAIL-R2 in the breast cancer cells [238]. This finding demonstrates that drug effects on cancer cells are likely tissue type-dependent, and emphasizes the importance of evaluating cytokine induction in the tissue of interest. Treatment of CRC cells with clinically relevant targeted treatments revealed several effects that were observed across drug treatments despite highly heterogeneous cell line mutational profiles and therapeutic mechanisms of action. These effects included downregulation of VEGF, CXCL9, and IL-8, and included upregulation of CXCL14, CCL5, and CXCL5. Other effects were more drug-class dependent [239]. Together, studies like these could support the logical combination of treatments to enhance or direct the immune system towards anti-tumor immunity in CRC.

Another area of great importance is the development and validation of predictive biomarkers of response to IT in CRC. While neoantigen load, level of tumor-infiltrating CD8+ T cells, and PD-1/PD-L1 expression are relatively reliable biomarkers, they have limited sensitivity and specificity, and are currently limited to predicting responses only to ICB. It is reasonable to predict correlation between levels of certain biomarkers and response to specific treatments based on the overarching mechanism of specific ITs. For example, it is likely that positive responses to ITs that are ultimately dependent on the action of CD8+ cytotoxic T cells (such as ICB, certain adoptive cell transfers, and cancer vaccinations) can be predicted based on the level of CD8+ T cells in the tumor. Similarly, it is reasonable to assume that response to targeted therapies depend on the expression level of that target, that NK cell-based therapies are more effective in patients who maintain high levels of NK cells, and so on. However, validation of the majority of these biomarkers remains to be completed.

Other limitations of IT include cost and development of resistance. Though insurance companies may pay for FDA-approved drugs, patient copays can be very high and can limit access to certain expensive drugs [240]. Resistance mechanisms to IT in CRC inevitably develop. Of highest clinical relevance is common resistance mechanisms to anti-PD-1/PD-L1 and anti-CTLA-4 ICB, which include downregulation of DC recruitment, insufficient activated or tumor-specific T cells, downregulation of antigen presentation, downregulation of IFN- γ, exhaustion of TILs, and increased immunosuppressive cells or cytokines. New therapeutic targets and combinations are actively being investigated to overcome these resistance mechanisms [241]. Many strategies are in clinical trials, including combination treatment with multiple ICBs, combination of ICB with costimulatory checkpoint molecules, priming cold tumors with cancer vaccines, ACT or oncolytic virus treatment before or concurrently with ICB, or modulating the microbiome to increase CRC-specific responses to ICB [135].

In conclusion, IT holds great promise to improve the lives of CRC patients. ICB treatments have been given the most attention in recent years, likely due to their recent FDA approval. ICB response rates in a majority of CRC patients are very low, necessitating the investigation of other types of IT, combination treatments, and predictive biomarkers. The results of ongoing trials that are investigating these topics are highly anticipated.

## Figures and Tables

**Figure 1 cancers-14-01028-f001:**
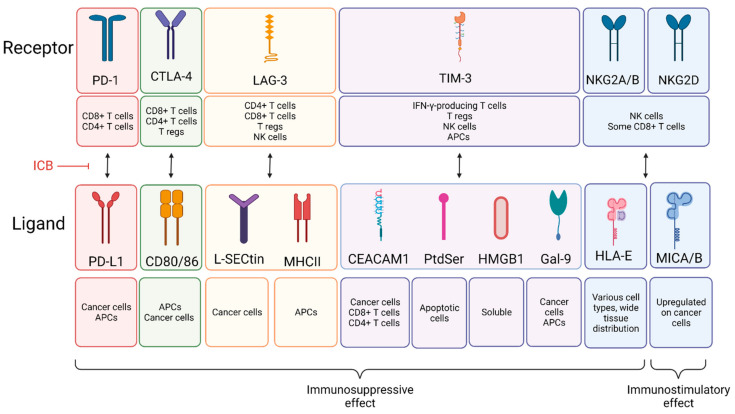
Immune cell and cancer cell interactions relevant to ICB therapy. CD8+ T cells, CD4+ T cells, NK cells, and T regs express receptors that are susceptible to binding by various cognate ligands present on the surface of colorectal cancer cells or APCs. Ligand binding results in immunosuppressive, or immunostimulatory in the case of NKG2D, signaling. ICB is a therapeutic approach involving inhibition of various receptor–ligand interactions. APC, antigen-presenting cell; ICB, immune checkpoint blockade; NK cell, natural killer cell. Created in BioRender.

**Figure 2 cancers-14-01028-f002:**
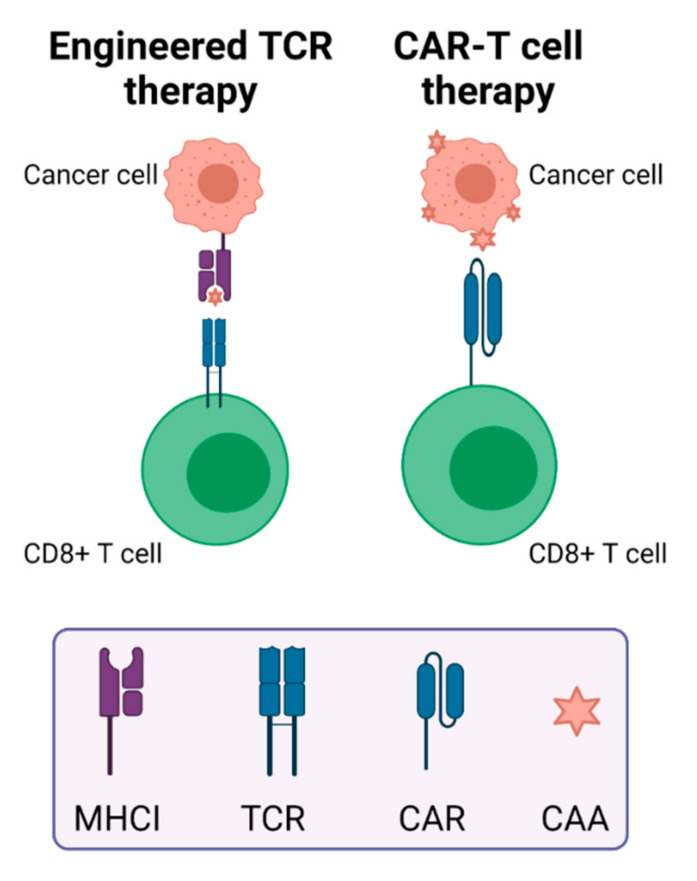
Engineered TCR therapy vs. CAR-T cell therapy. Engineered TCR T cell therapy involves engineering T cells with receptors that are specific for MHC class I-presented antigenic peptides. CAR-T cell therapy involves direct recognition of CAAs on the cancer cell surface. CAA, cancer-associated antigen; CAR, chimeric antigen receptor; MHCI, major histocompatibility complex class I; TCR, T cell receptor. Created in BioRender.

**Figure 3 cancers-14-01028-f003:**
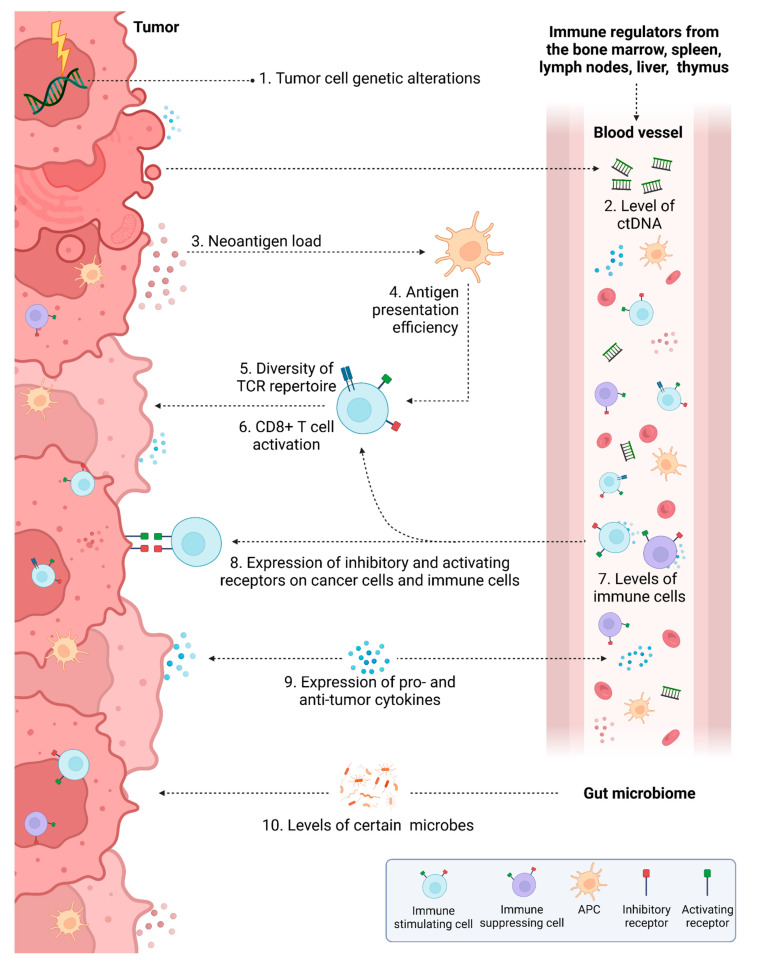
Key biomarkers of response to IT in CRC. (1) Certain genetic alterations in CRC cells have been associated with better response to IT. (2) A high level of ctDNA correlates with worse response to many cancer therapies including IT. (3) Neoantigen load is a key biomarker of response to several ITs including ICB. (4) High antigen presentation efficiency is associated with better response to T cell-based ITs. (5) Diversity of the TCR repertoire is associated with better response to IT. (6) A high level of CD8+ T cell activation is associated with better response to different ITs. (7) Levels of circulating and tumor-infiltrating immune cells including immune stimulatory cells (such as T cells and NK cells) and immune suppressor cells (such as T regs, MDSCs, and M2 macrophages), predict response to many types of IT. (8) Expression of inhibitory and activating receptors is impacts response to many types of IT, especially ICB. Circulating exosomes containing soluble receptors may also have predictive value. (9) Expression of pro- and anti-tumor cytokines by cancer cells and immune cells plays a role in response to all ITs. (10) Levels of certain gut microbes in the tumor, and levels of circulating microbe-associated compounds, may predict response to IT. APC, antigen-presenting cell; CRC, colorectal cancer; ctDNA, circulating tumor DNA; ICB, immune checkpoint blockade; IT, immunotherapy; MDSC, myeloid-derived suppressor cell; NK, natural killer; TCR. T cell receptor; T reg, T regulatory cell.

**Table 1 cancers-14-01028-t001:** The molecular action and cellular expression of targets relevant to IT in CRC.

Target	Expressed by	Molecular Action	Ref.
PD-1	CD8+ T cells, CD4+ T cells	Suppresses CD8/4+ T cell activity	[11,12]
PD-L1	Cancer cells, APCs	Suppresses CD8/4+ T cell activity	[11,12]
CTLA-4	CD8+ T cells, CD4+ T cells, T regs	Suppresses CD8/4+ T cell activity Enhances T reg activity	[13]
LAG-3	CD8+ T cells, CD4+ T cells, T regs, NK cells	Suppresses CD8/4+ T cell activity Enhances T reg activity	[14]
TIM-3	IFN-γ-producing T cells, T regs, NK cells, APCs	Suppresses T cell, NK cell, and APC activity May enhance T reg activity	[14]
NKG2	NK cells, some CD8+ T cells	NKG1A/B: Suppresses NK cell activity NKG2D: Enhances NK cell activity	[15,16]
EpCAM/CD326	Cancer cells	Upregulates oncogene expression and cell proliferation	[17]
EGFR	Cancer cells	Triggers cell proliferation	[18]
VEGF	Cancer cells	Stimulates angiogenesis	[18]
CD133	Cancer stem cells	Play a role in chemotherapy resistance	[19]
HER3	Cancer cells	Promotes cell proliferation	[20,21]
HER2	Cancer cells	Promotes cell proliferation	[22,23]
CD3	CD8+ T cells, CD4+ T cells	Used in bsTCEs to engage T cells	[24]
GPA33	Cancer cells	Function unclear; overexpressed in CRC	[25,26]
GUCY2C	Cancer cells	Maintains intestinal homeostasis	[27]
CEA	Cancer cells	May inhibit cell differentiation, apoptosis, and anoikis	[28]
Mutant KRAS	Cancer cells	Mutation causes overactive cell proliferation	[29]
Mutant TP53	Cancer cells	Mutation causes loss of tumor suppressive ability and possible gain of oncogenic properties	[30]
Angiopoietin-2	Cancer cells	Stimulates angiogenesis	[31]
MET	Cancer cells	Promotes cellular proliferation, motility, migration and invasion	[32,33]
DR5	Cancer cells	Induces apoptosis	[34]
EphA2	Cancer cells	Controls cell–cell repulsion or adhesion	[35]
MUC-1	Cancer cells	Associated with invasion, metastases	[36]
Survivin	Cancer cells	Inhibits cell death	[37,38]
SART3	Cancer cells	Spliceosome recycling factor; RNA-binding protein; overexpressed in CRC	[39,40]
IL-2	Cancer cells, CD4+ T cells	Stimulates T cells and NK cells	[41,42]
IL-12	APCs	Stimulates T cells and NK cells	[41]
IL-11	Epithelial cells, endothelial cells, fibroblasts, myeloid cells	Tumor-promoting cytokine	[41,43]
IL-6	Epithelial cells, myeloid cells	Tumor-promoting cytokine	[41]
IL-1α	Epithelial cells, myeloid cells	Multi-functional: promotes inflammatory carcinogenesis; promotes antitumour immunity	[41,44,45]
IL-1β	Epithelial cells, myeloid cells	Multi-functional: promotes inflammation-induced carcinogenesis; recruits antineoplastic cells, may block metastatic outgrowth	[41,44,45]
IL-33	Epithelial cells, endothelial cells, adipocytes, fibroblasts, DCs	Recruits, activates, and degranulates eosinophils	[46,47]
IFN-γ	Cancer cells, CD4+ T cells, CD8+ T cells, γδ T cells, NK cells	Likely a tumor-inhibiting cytokine	[48]
STING	Widely expressed in immune and non-immune cells	Stimulates IFN genes and cellular senescence	[49,50]
CCR5	Cancer cells, MDSCs, T regs, monocytes, macrophages, DCs, Th1 cells, activated T cells, NK cells	Enhances cancer cell motility; enhances MDSC and T reg infiltration	[51,52]

CRC, colorectal cancer; IT, immunotherapy.

**Table 2 cancers-14-01028-t002:** Advantages and disadvantages of the IT approaches for treatment of CRC.

Immunotherapy	Advantages	Disadvantages
Immune checkpoint blockade	Sensitive, specific, additional T cell activation mechanisms possible, can be combined with each other	Most are mAbs and confer the same disadvantages, systemic toxicity is likely, response rates are low in CRC
Adoptive cell therapy	Personalized, permanent T cell modification confers immune memory	Expensive, difficult to manufacture, GvHD, CRS and B cell aplasia common
Monoclonal antibody	Relatively inexpensive, specific, effective across cancer types, can be conjugated easily	Target identification expensive, inadequate pharmacokinetics and tissue accessibility, resistance development common
Oncolytic virus therapy	Specific to cancer cells, may prime immune system to boost response to other ITs	Anti-viral immunity may reduce efficacy
Cancer vaccines	Specific, can be personalized	Rejection is possible due to delivery of foreign antigens
Immune system modulators	Many are FDA-approved, small size facilitates access to cancer cells, relatively inexpensive, can stimulate general anti-cancer immune response	Low specificity possible, risk of immediate onset CRS (cytokine storm]

Adapted from [56]. CRC, colorectal cancer; CRS, cytokine release syndrome; IT, immunotherapy; GvHD, graft versus host disease.

**Table 3 cancers-14-01028-t003:** The number of clinical trials involving various types of immunotherapy for the treatment of colorectal cancer.

Number of Clinical Trials											
	ICB	ACT	mAb	Conj. Ab	bsAb	Virus	Vaccine	IFN	IL	IMiD	STING
Completed	31	8	67	3	3	3	46	11	7	2	0
Active, not recruiting	46	6	19	-	-	1	4	1	4	-	0
Recruiting	111	15	52	2	8	2	18	4	5	-	1
Not yet recruiting	16	7	9	1	3	-	2	1	-	-	0
Terminated	11	4	29	1	1	1	11	2	5	1	0
Withdrawn	10	2	12	2	2	1	5	1	2	-	0
Suspended	-	1	-	-	-	-	1	-	-	-	0
Unknown	9	14	15	-	2	-	9	5	2	3	0
Total	234	57	203	9	19	8	96	25	25	6	1
Search terms											
ICB	immune checkpoint blockade OR checkpoint blockade OR immune checkpoint inhibitor OR anti-PD-1 OR anti-pdl1 OR anti-ctla4 OR anti-lag3 OR anti-tim3 OR anti-nkg2 OR PD-1 OR pdl1 OR ctla4 OR lag3 OR tim3 OR nkg2										
ACT	adoptive cell therapy OR adoptive cell transfer OR cellular adoptive immunotherapy OR t cell transfer therapy OR tumor-infiltrating lymphocyte OR engineered t cell OR t cell receptor therapy OR car t cell OR NK cell										
mAb	monoclonal antibody OR monoclonal antibodies										
Conj. Ab	conjugated antibody OR conjugated antibodies										
bsAb	bispecific antibody OR bispecific antibodies										
Virus	oncolytic virus										
Vaccine	vaccine										
IFN	interferon										
IL	interleukin										
IMiD	thalidomide										
STING	sting										

ClinicalTrials.gov search was conducted as follows: condition or disease = colorectal cancer; intervention/treatment = search terms above; study type = interventional. Search was performed on 30 October 2021. Some trials fall into two groups, for example many ICB therapies are mAb-based. Clinical trials involving combination treatments also may fall into two groups. ACT, adoptive cell therapy; bsAb, bispecific antibody; conj. Ab, conjugated antibody; ICB, immune checkpoint blockade; IFN, interferon; IL, interleukin; IMiD, immunomodulatory drug; mAb, monoclonal antibody.

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
