# Peer review of "Immunotherapy for Colorectal Cancer: Mechanisms and Predictive Biomarkers"

_cancers, 2022, doi:10.3390/cancers14041028_

Round 1

Reviewer 1 Report

The approach to reveal mechanisms and predictive biomarkers is useful for patients with colorectal cancer who would undergo immunotherapy. All physicians and scientists recognize the important issue.

Authors comprehensively reviewed cutting edge of immunotherapies including experimental therapies as well as on-going clinical trials, followed by predictive biomarkers for response to these immunotherapies. The review is enough as a proceeding report, on the other hand, a review article requires for originality and priority to recognize the evidence.

There are limited data to conclude the efficacy of various immunotherapies in colorectal cancer except for immune checkpoint blockade and monoclonal antibodies; therefore, it is hard to reveal the biomarkers among the candidate factors in variety of immunotherapies. 

Authors need to explain the correlations between immunotherapies and predictive biomarkers in the session as Discussion and open questions. To answer the research questions, it is better to describe profound mechanisms underlying the efficacy of the immunotherapies.

Authors cited recent references, abstracts and database; however, pivotal clinical trials and novel research articles need to be cited for review.

Figures 1 and 2 are ordinary like a book chapter, therefore it is better to describe in relation to predictive biomarkers. 

Author Response

Reviewer 1:

The approach to reveal mechanisms and predictive biomarkers is useful for patients with colorectal cancer who would undergo immunotherapy. All physicians and scientists recognize the important issue.

Authors comprehensively reviewed cutting edge of immunotherapies including experimental therapies as well as on-going clinical trials, followed by predictive biomarkers for response to these immunotherapies. The review is enough as a proceeding report, on the other hand, a review article requires for originality and priority to recognize the evidence. There are limited data to conclude the efficacy of various immunotherapies in colorectal cancer except for immune checkpoint blockade and monoclonal antibodies; therefore, it is hard to reveal the biomarkers among the candidate factors in variety of immunotherapies.
We thank the reviewer for their comments and agree that the current list of biomarkers for response to IT in CRC is not exhaustive or well-defined. We have added a Figure 3 to help review the biomarkers for different types of immunotherapies, based on current understanding.

Authors need to explain the correlations between immunotherapies and predictive biomarkers in the session as Discussion and open questions. To answer the research questions, it is better to describe profound mechanisms underlying the efficacy of the immunotherapies.
Additional text was added to section 5 “Discussion & open questions” to explain the correlation between the overarching mechanism of each IT and its biomarkers of response.

Authors cited recent references, abstracts and database; however, pivotal clinical trials and novel research articles need to be cited for review.
Additional pivotal clinical trials have been added including KEYNOTE 028 and CheckMate 142. References were added where necessary for all clinical trials and novel research articles.

Figures 1 and 2 are ordinary like a book chapter, therefore it is better to describe in relation to predictive biomarkers.
We agree with the reviewer that Figures 1 and 2 are more mechanistic but believe they are important to include for a foundational understanding of the topics discussed. We have added Figure 3, which helps visually integrate the information about biomarkers.

Reviewer 2 Report

In the present review, the authors discuss the present preclinical developments in the area of immunotherapy for patients with colorectal cancer. I have several reservations, my comments are appended as below:

Major comments:

  1. Chemotherapy- is it the same for all pathological stages? (line 49-51)
  2. Is MSI follows the pathological stages or independent of it?
  3. Authors should be consistent in referring to the abbreviations: IT/ CPI
  4. Authors should discuss the potential toxicities of presently approved therapies.
  5. Authors may add a table representing the cells on which the receptors are expressed and molecular action in brief.
  6. Make sure to annotate every sentence with competent reference. For instance, lines 167-170.
  7. Line 171-173- share details on cell types expressing these ligands.
  8. While referring to clinical studies with patients, authors should include the statistical inference (HR, p-value). For instance, lines 223, 242, etc

9. In addition to presently discussed factors, other cofounders are also discussed in existing literature PMID: 33076303, authors may refer and add a para.  

  1. Engineered T cell therapy: how the costimulatory signal is provided?
  2. while discussing the factors authors should try to extrapolate its reported/predicted use to colorectal cancer. For instance, section Immune System Modulators, PD-L1 expression.

Author Response

Reviewer 2

In the present review, the authors discuss the present preclinical developments in the area of immunotherapy for patients with colorectal cancer. I have several reservations, my comments are appended as below:

Major comments:

  1. Chemotherapy- is it the same for all pathological stages? (line 49-51)
    We thank the reviewer for their comments and have clarified common chemotherapy regimens across CRC pathological stages in the first paragraph of the Introduction.

  2. Is MSI follows the pathological stages or independent of it?
    The following text was added to the second paragraph of the Introduction to clarify this point: “MSI status is generally independent of pathological stage.”

  3. Authors should be consistent in referring to the abbreviations: IT/ CPI
    Consistent use of the abbreviations IT (referring to immunotherapy) and ICB (immune checkpoint blockade was confirmed. We did not use CPI or other acronyms to describe ICB such as ICI.
    IT refers to all types of immunotherapy, whereas ICB refers only to types of therapy that involve immune checkpoint inhibition. This is described on lines 69-71: “Different types of IT include immune checkpoint blockade (ICB), adoptive cell therapies (ACT), monoclonal antibodies (mAbs), oncolytic viruses, anti-cancer vaccines, and immune system modulators.” This sentence was modified slightly to clarify the difference between these abbreviations.

  4. Authors should discuss the potential toxicities of presently approved therapies.
    Toxicities of presently approved therapies were added.

  5. Authors may add a table representing the cells on which the receptors are expressed and molecular action in brief.
    This information is now included as Table 1.

  6. Make sure to annotate every sentence with competent reference. For instance, lines 167-170.
    References were added to lines 167-170 (lines 240-243 in the revised version). The rest of the text was reviewed and any missing references were added.

  7. Line 171-173- share details on cell types expressing these ligands.
    The cell types expressing these ligands were added to the text.

  8. While referring to clinical studies with patients, authors should include the statistical inference (HR, p-value). For instance, lines 223, 242, etc
    HR and p-values were added to any text that referred to clinical studies that are completed and have statistical information available.
    Line 223 does not refer to a specific clinical trial, and line 242 references a clinical trial that does not have results available yet.

  9. In addition to presently discussed factors, other cofounders are also discussed in existing literature PMID: 33076303, authors may refer and add a para.  
    An additional section 3.4 “Other factors influencing response to IT” was added, along with the suggested reference.

      10.Engineered T cell therapy: how the costimulatory signal is provided?
           Costimulatory molecules are now mentioned in the text on lines 328-329.

      11. While discussing the factors authors should try to extrapolate its        reported/predicted use to colorectal cancer. For instance, section Immune System Modulators, PD-L1 expression.
The effect of PD-L1 expression on the predicted utility of immune system modulators was added.

We agree with the reviewer that the current list of biomarkers for response to IT in CRC is not exhaustive or well-defined. We predict that certain biomarkers in development will have predictive value for other types of immunotherapy that have similar overarching mechanisms. This is now discussed on lines 714-717 and lines 1080-1088.

Reviewer 3 Report

Thank you for the invitation to review the Immunotherapy for Colorectal Cancer: Mechanisms and Predictive Biomarkers manuscript. The review article is a comprehensive literature search highlighting the progress in understanding colorectal cancer treatment, a rapidly expanding field in treating this devastating cancer. The manuscript is a well-written review, starting with a precise aim and detailing several related sections. It is not easy to write a review paper considering such a highly complex disease. Authors have performed an admirable search, collecting the relevant information condensing them in a short paragraph of considerable value. The graphics are excellent, detailed descriptions of the content in legend and the texts helping the reader to follow the report without losing interest or getting confused. The manuscript provides detailed information that helps medical scientists and clinicians manage cancer patients. A review manuscript is not just a collection of data and summarising pieces of literature in short paragraphs. The review articles must contain authors reflections on the strength and shortfall of the related literature in an objective manner. Immunotherapy at best works for 50 per cent of cancer patients and eventually develops other complications such as chronic pleuritis. What will the authors recommend to the oncologist team or respiratory physician treating such patients? Let the patient die, continuing with drainage of the accumulated fluid in the lung. It is of high value that the authors objectively reevaluate each section and describe the advantages and disadvantages of the immunotherapy method (s) used. It is beneficial to the readers if the authors summarise all the different immunotherapy methods strengths and shortfalls and the author's recommendation a table. Please provide a list of abbreviations used in the manuscript; it is rather tiring to go back and forth to remember rach abbreviations. Overall, a manuscript of significance will add to our knowledge and advance the immunotherapeutic methods of treatment of colorectal cancer. 

Author Response

Thank you for the invitation to review the Immunotherapy for Colorectal Cancer: Mechanisms and Predictive Biomarkers manuscript. The review article is a comprehensive literature search highlighting the progress in understanding colorectal cancer treatment, a rapidly expanding field in treating this devastating cancer. The manuscript is a well-written review, starting with a precise aim and detailing several related sections. It is not easy to write a review paper considering such a highly complex disease. Authors have performed an admirable search, collecting the relevant information condensing them in a short paragraph of considerable value. The graphics are excellent, detailed descriptions of the content in legend and the texts helping the reader to follow the report without losing interest or getting confused. The manuscript provides detailed information that helps medical scientists and clinicians manage cancer patients.

A review manuscript is not just a collection of data and summarising pieces of literature in short paragraphs. The review articles must contain authors reflections on the strength and shortfall of the related literature in an objective manner.
We thank the reviewer for their comments and have better addressed the strengths and weakness of IT strategies in the text and in Table 2.

Immunotherapy at best works for 50 per cent of cancer patients and eventually develops other complications such as chronic pleuritis. What will the authors recommend to the oncologist team or respiratory physician treating such patients? Let the patient die, continuing with drainage of the accumulated fluid in the lung.
We agree that it is important to highlight the low IT response rate in CRC, and have emphasized this in the Introduction. As far as chronic pleuritis/pulmonary toxicity, this side effect is largely found in lung cancer and melanoma patients who receive IT and does not seem to be a significant issue for colorectal cancer patients receiving IT (doi:10.3390/cancers11030305). Therefore, this side effect may not be relevant to address in this Review, which focuses on colorectal cancer.
It is of high value that the authors objectively reevaluate each section and describe the advantages and disadvantages of the immunotherapy method (s) used.
We agree with the Reviewer and have included Table 2, which lists the pros and cons of each IT strategy, and have emphasized these points in the text.
It is beneficial to the readers if the authors summarise all the different immunotherapy methods strengths and shortfalls and the author's recommendation a table.
We agree with the Reviewer and have included Table 2, which lists the pros and cons of each IT strategy, and have emphasized these points in the text.
Please provide a list of abbreviations used in the manuscript; it is rather tiring to go back and forth to remember rach abbreviations.
A list of abbreviations has been included after the Discussion & open questions section.

Overall, a manuscript of significance will add to our knowledge and advance the immunotherapeutic methods of treatment of colorectal cancer.

Additional revisions made by the Authors:

  1. Repeated definitions of abbreviations were removed.
  2. Abbreviation lists at the end of Table and Figure legends were made alphabetical.
  3. Some minor typos were corrected.
  4. Detailed description of FLT3L study was moved from the Discussion and into subsection 2.1.1, where it is first mentioned.
  5. Additional Figures and Tables based on Reviewer comments have been cited accordingly in the text.
  6. Additional subsection 3.3.3 “cytokines” was added.
  7. When referring to microsatellite status, abbreviations were made consistent (used MSI/MSS throughout).

Round 2

Reviewer 1 Report

Authors elegantly revised the manuscript adding Table 1 and Figure 3. This revised version would be highly expected level of review article for physicians and researchers.

I would like to suggest to the Authors that PD-L1 is expressed on APCs other than cancer cells in Table 1 and Figure 1. Please add APCs in the column.

Author Response

Revisions (Round 2) for “Immunotherapy for Colorectal Cancer: Mechanisms and Predictive Biomarkers”

Lindsey Carlsen, Kelsey E. Huntington, and Wafik S. El-Deiry

Author responses to Reviewer comments in this cover letter are in red text. All modifications of the manuscript were made using track changes.

Reviewer 1:

Authors elegantly revised the manuscript adding Table 1 and Figure 3. This revised version would be highly expected level of review article for physicians and researchers.

I would like to suggest to the Authors that PD-L1 is expressed on APCs other than cancer cells in Table 1 and Figure 1. Please add APCs in the column.
We thank the reviewer for their comments and suggestion. Expression of PD-L1 by APCs has been added to Table 1 and Figure 1.

Reviewer 2 Report

I congratulate the authors for providing the modifications, with that, the manuscript is closer to publication. I however recommend taking care of a few minor points:

  1. It would be great if authors annotate table 1 with competent reference(s) like they do for table 2.
  2. Section 3.2.6, section 3.3.2- reference 25- annotate with a number of patients and statistical inference.

Author Response

Revisions (Round 2) for “Immunotherapy for Colorectal Cancer: Mechanisms and Predictive Biomarkers”

Lindsey Carlsen, Kelsey E. Huntington, and Wafik S. El-Deiry

Author responses to Reviewer comments in this cover letter are in red text. All modifications of the manuscript were made using track changes.

Reviewer 2:

I congratulate the authors for providing the modifications, with that, the manuscript is closer to publication. I however recommend taking care of a few minor points:

  1. It would be great if authors annotate table 1 with competent reference(s) like they do for table 2.
    We thank the reviewer for their comments and agree with their suggestions. References have been added to Table 1 along with other minor additional information.

  2. Section 3.2.6, section 3.3.2- reference 25- annotate with a number of patients and statistical inference.
    Number of patients and statistical inference were added to reference 181 in section 3.2.6.
    Number of patients and additional relevant detail were added to reference 25 in section 3.3.2. Statistical inference is not available from this source.

Other revisions made by the Authors:

  • Minor improvements to sentence structure were made throughout the text.